# Published patterns of spin in biomedical literature: a protocol for a meta-research study

Naichuan Su [1], Michiel van der Linden,[1] Geert JMG van der Heijden,[1] Stefan Listl,[2,3] Stefan Schandelmaier [4,5] Clovis M Faggion Jr[6]

For numbered affiliations see end of article.

**Correspondence to**
Dr Naichuan Su; n.su@acta.nl

## ABSTRACT

**Introduction** Spin is defined as reporting practices that distort the interpretation of results and create misleading conclusions by suggesting more favourable results. Such unjustifiable and misleading misrepresentation may negatively influence the development of further studies, clinical practice and healthcare policies. Spin manifests in various patterns in different sections of publications (titles, abstracts and main texts). The primary aim of this study is to identify reported spin patterns and assess the prevalence of spin in general, and the prevalence of spin patterns reported in biomedical literature based on previously published systematic reviews and literature reviews on spin.

**Methods and analysis** PubMed, EMBASE and SCOPUS will be searched to identify systematic or literature reviews on spin in biomedicine. To improve the comprehensiveness of the search, the snowballing method will be used to broaden the search. The data on spin-related outcomes and characteristics of the included studies will be extracted. The methodological quality of the included studies will be assessed with selective items of the A MeaSurement Tool to Assess systematic Reviews-2 checklist. A new classification scheme for spin patterns will be developed if the classifications of spin patterns identified in the included studies vary. The prevalence of spin and spin patterns will be pooled based on meta-analyses if the classification schemes for spin are comparable across included studies. Otherwise, the prevalence will be described qualitatively. The seriousness of spin patterns will be assessed based on a Delphi consensus study.

**Ethics and dissemination** This study has been approved by the Academic Centre for Dentistry Amsterdam Ethics Review Committee (2020250). The study will be submitted to a peer-reviewed scientific journal.

**Registration** Open Science Framework: osf.io/hzv6e

## Strengths and limitations of this study

► While previous systematic or literature reviews on spin concerned the identification of spin patterns in various study designs, this will be the first systematic review focussing on the prevalence of spin patterns and seriousness of the impact of spin patterns on research and clinical practice based on the previously published systematic or literature reviews.

► This study may give insight into which spin patterns occur most frequently and may cause most serious consequences in research and clinical practice, and thus should be given most attention when researchers, clinicians and policymakers read and write scientific papers.

► Potential limitation may be that the various definitions of spin and the various classification schemes for spin patterns used in the included studies may impact the pooling and interpretation of the results in this study.

► Limiting the search to literature reviews or systematic reviews only is a potential limitation of the study because some meta-studies may not be reviews.

► Applying the A MeaSurement Tool to Assess systematic Reviews-2 (AMSTAR-2) checklist to all the included studies for methodological quality assessment is another potential limitation because some of the included studies may have other designs than systematic reviews.

## INTRODUCTION

Spin is defined as intentionally or unintentionally inaccurate, unfair or partial reporting practices that distort the interpretation of research and create misleading conclusions by suggesting more favourable results.[1] It can be hypothesised that such unjustifiable and misleading misrepresentation may flatter research and make it more attractive for publication and citation and consequently may receive unwarrantably higher impact scores.

Spin is reported in 57% of the published clinical trials on average based on a systematic review on spin.[1] Based on the hypothesis, spin may lead to overpromising and misleading information in transfer of knowledge, introduce misconceptions to researchers, clinicians and policymakers, and stimulate ineffective or even harmful financial investments from funding organisations and health institutions. Therefore, spin has a negative impact on advancing healthcare practice and population health, adversely influences health policy planning, adds to research

waste, decreases the reproducibility of research, hampers the progress of science and reduces return-of-investment from research.[1–3]

Spin manifests in different sections of publications (titles, abstracts and main texts) and in various patterns. For example, the title of a publication may claim a beneficial effect of an experimental intervention which is not supported by the reported findings, or the conclusion of a publication is not supported by the reported findings, or the provided recommendations for clinical practice are not in line with the study conclusion. Those spin patterns can be classified into three main forms: misleading reporting, misleading interpretation and inappropriate extrapolation.[4 5] The misleading reporting was defined as incomplete or inadequate reporting of the methods, study analysis, study results or any important information that could be misleading to the readers, such as the selective reporting of or overemphasis on statistically significant secondary outcomes but ignoring the statistically non-significant primary outcomes. The misleading interpretation was defined as an interpretation of the study results that could be misleading to the readers, such as the conclusion claiming equivalence or comparable effectiveness for non-significant results. The inappropriate extrapolation was defined as an inappropriate generalisation of study results, such as the conclusion claiming the recommendation to use the treatment in clinical practice in an observational study.[1 4]

Several studies on the prevalence of spin and the impact of spin on research in general have been published. For example, multiple systematic reviews or literature reviews investigated spin in primary studies such as randomised controlled trials (RCTs), non-RCTs, diagnostic accuracy studies, prognostic studies or systematic reviews performed in different clinical areas.[2 5–8] In addition, a meta-research study on spin was published in 2019, which aimed to identify spin and patterns of spin in the studies on spin.[9] Almost all these previous studies on spin showed that spin was very common across different types of the primary studies, ironically even in the studies on spin.[9] In addition, a study reported that spin in abstracts may influence clinicians' decision-making on the treatment effect in the field of cancer,[10] while another study reported that spin in health news stories reporting studies of pharmacologic treatment may influence both patients' and caregivers' decision-making on the treatment effect.[11] Another meta-research study on spin was published in 2016.[1] It focused on the theory of spin and the overall prevalence of spin in primary studies based on systematic or literature reviews on spin. These studies have revealed that spin is highly frequent in medical publications and can significantly impact both clinicians' and patients' decision-making in clinical practice. The main aim of these studies was to define, describe and understand spin. From the publications on spin available to date, different patterns of spin seem to emerge, but a comprehensive overview of spin patterns is lacking. Spin patterns vary in the prevalence in research and in the consequences on research. The ignorance on those spin patterns with the highest prevalence and the most serious impact on research may widely and severely harm the reliability, transparency and accuracy of translation, dissemination and implementation of the evidence from medical and health research to practice. Identification of the prevalence and the impact of spin patterns will help increase the awareness of researchers, clinicians, reviewers and policymakers on which spin patterns should be given most attention when they read, review and write scientific papers. Besides, to develop the practical and targeted guidance on the identification and prevention of spin in research, the manifestations of spin should be known. The specific and targeted guidance on how to identify and prevent such spin patterns in research can be made accordingly to help prevent spin in the future publications and prevent the use and transfer of the spinned evidence in research. To date, however, meta-research studies on the prevalence of spin patterns in biomedical publications and seriousness of the impact of different spin patterns on research and clinical practice are scarce.

Therefore, the aims of the study are to: (1) identify reported spin patterns and assess the prevalence of spin in general and the prevalence of spin patterns reported in biomedical literature based on previously published systematic reviews and literature reviews on spin; and (2) assess the seriousness of the impact of spin patterns on research and clinical practice. Based on the spin patterns found in the current study which are most frequent in research or have the most serious consequences on research, we will derive recommendations for researchers, clinicians and policymakers to prevent such spin patterns in future research.

## METHODS AND ANALYSIS
The present protocol has been registered in the Open Science Framework and reported in accordance with the Preferred Reporting Items for Systematic Review and Meta-analysis Protocols statement.[12] The present protocol has been approved by Academic Centre for Dentistry Amsterdam Ethics Review Committee (2020250).

### Search strategy
Relevant publications will be searched in electronic bibliographic sources, including PubMed, EMBASE and SCOPUS. The full search strategies, which have been modified by a senior librarian from Vrije Universiteit Amsterdam, the Netherlands (see acknowledgement), are described in box 1. Searches will be limited from the last 10 years (from 2010), as from the first publication to identify spin in clinical trials.[6] The searching for the abstracts is planned to be done in early May 2021.

**Box 1    Search strategies in PubMed, EMBASE and SCOPUS**

**PubMed**
#11 Search: #10 AND ("2010"[Date-Publication]: "3000" [Date-Publication])
#10 Search: #8 OR #9
#9 Search: #7 AND systematic[(sb)]
#8 Search: #7 AND review[(Publication Type)]
#7 Search: #3 OR #4 OR #5 OR #6
#6  Search: ("misinterpretation"[(tiab))  OR  "overinterpretation"[(tiab)])
AND
"result*"[(tiab)]
#5 Search: "distorted results"[(tiab)] OR "distorted reporting"[(tiab)] OR
"distorted presentation"[(tiab)] OR "distorted interpretation"[(tiab)]
#4 Search: "reporting bias"[(tiab)] OR "interpretation bias"[(tiab)]
#3 Search: #1 AND #2
**#2** Search: "Publication Bias"[(Mesh)] OR "report*"[(tiab)] OR "bias*"[(-tiab)] OR
"publish*"[(tiab)] OR "non-significant*"[(tiab)] OR
"nonsignificant*"[(tiab)]
#1 Search: "Spin"[(tiab)]

**EMBASE**
#11 #10 AND [2010–2020]/py
#10 #8 OR #9
#9 #7 AND 'systematic review*':ti,ab,kw
#8 #7 AND [(review)]/lim
#7 #3 OR #4 OR #5 OR #6
#6 (('misinterpretation' OR 'overinterpretation') NEAR/4 'result*'):ti,ab,kw
#5 distorted results':ti,ab,kw OR 'distorted reporting':ti,ab,kw OR 'distorted presentation':ti,ab,kw OR 'distorted interpretation':ti,ab,kw
#4 'reporting bias'/exp OR 'interpretation bias'/exp
#3 #1 AND #2
#2 'publication bias'/exp OR 'report*':ti,ab,kw OR 'bias*':ti,ab,kw OR 'publish*':ti,ab,kw OR 'non-significant*':ti,ab,kw OR 'nonsignificant*':ti,ab,kw
#1 'spin':ti,ab,kw

**SCOPUS**
#10 #9 AND (PUBYEAR >2009)
#9 #6 OR #8
#8 #4 AND #7
#7 TITLE-ABS-KEY ("systematic review*")
#6 #4 OR #5
#5 DOCTYPE (re)
#4 #1 OR #2 OR #3
#3 TITLE-ABS-KEY (("misinterpretation" OR "overinterpretation")
W/3 ("result*"))
#2 TITLE-ABS-KEY ("reporting bias" OR "interpretation bias" OR
"distorted results" OR "distorted reporting" OR "distorted presentation "
OR "distorted interpretation")
#1 TITLE-ABS-KEY (("spin") W/3 ("report*" OR "bias*" OR
"publish*" OR "non-significant*" OR "nonsignificant*"))

To improve the comprehensiveness of the search and identify potentially eligible studies which are not retrieved from the databases, the snowballing method will be used to broaden the search.[13] That is, we inspect the references that are cited in the studies eligible for inclusion (backward citation searching).[14] Then, we use 'see all similar articles' function at PubMed to check the first 20 most 'similar studies' for each eligible publication retrieved from either the databases or the lists of references of each eligible publication.

## Eligibility criteria
We will include publications that satisfy all of the following criteria:
1. They reported systematic or literature reviews on spin. That is, the publications aimed to examine spin for the primary studies in a specific or broad field of medicine. The review articles are defined as a more or less systematic way of collecting and synthesising previous research.[15] A review will be defined as systematic when authors of the review made clear the intention of performing a systematic review. If a publication met the definition based on its adopted methods without using the word 'review' in the titles, abstracts or main texts, it will also be included.
2. They assessed spin of primary studies with any study design (eg, effect of treatment or prevention studies (including randomised and non-randomised trials) and diagnostic and prognostic accuracy studies, economic studies and systematic reviews) in any field of biomedical sciences.
3. They reported possible spin patterns and prevalence of spin or of spin patterns in the included primary studies.
4. They were published in the English language.

Two reviewers (NS and CMF) will select a sample (10%) of eligible studies to achieve good agreement (at least 80%) on inclusion and exclusion of publications, and thereafter the remainder selected by one reviewer (NS).[16] Full texts will be obtained for studies that meet the inclusion criteria, and whenever title and abstract provide insufficient information for inclusion.

## Data extraction
The characteristics of the included studies and spin-related outcomes will be extracted from each review directly to a standardised form. The standardised form will be developed in Microsoft Excel software.

The information on the characteristics of the included studies include: (a) publication year and first authors; (b) country/continent of the first author; (c) number of authors of the review; (d) type of review (systematic vs literature review); (e) medical specialty; (f) type of primary study evaluated in the review; (g) sponsorship of the review; (h) conflict of interest of authors of the review; (i) sampling methods used in the reviews.

The spin-related outcomes of the included studies include (a) in what sections the spin was assessed (eg, abstracts, methods, results and conclusions sections of the primary studies); (b) spin patterns identified in the reviews including labels, definitions and classification schemes; (c) prevalence of spin (overall and for different spin patterns) based on their original classification schemes.

Two reviewers (NS and CMF) will extract the data from a sample (10%) of eligible studies and achieve good

**Table 1** The critical and non-critical items of AMSTAR-2 that are applicable in this study

| Items | Response options |
|---|---|
| **Critical items** | |
| Item 4 Did the review authors use a comprehensive literature search strategy? | Yes/partial Yes/no |
| Item 7 Did the review authors provide a list of excluded studies and justify the exclusions? | Yes/partial Yes/no |
| Item 13 Did the review authors account for RoB in individual studies when interpreting/discussing the results of the review? | Yes/no |
| **Non-critical items** | |
| Item 5 Did the review authors perform study selection in duplicate? | Yes/no |
| Item 6 Did the review authors perform data extraction in duplicate? | Yes/no |
| Item 8 Did the review authors describe the included studies in adequate detail? | Yes/partial Yes/no |
| Item 10 Did the review authors report on the sources of funding for the studies included in the review? | Yes/no |
| Item 16 Did the review authors report any potential sources of conflict of interest, including any funding they received for conducting the review? | Yes/no |

AMSTAR, A MeaSurement Tool to Assess systematic Review.

agreement (at least 80%), with the remainder extracted by one reviewer (NS).[16]

## Assessment of methodological quality

The updated version of A MeaSurement Tool to Assess systematic Reviews (AMSTAR-2), which includes a total of 16 items, is mainly used to assess the methodological quality of systematic reviews that include randomised or non-randomised studies of healthcare interventions.[16] Because there is no validated tool to evaluate the methodological systematic or literature reviews so far, we will apply selective items of the AMSTAR-2 checklist[16] that fit the purpose of our research to assess the methodological quality of the included systematic or literature review on spin. Eight items from the AMSTAR-2 items are applicable and selected in the present study (table 1).

The response options to each item include 'Yes' and 'No'. For some items, 'Partial yes' is an additional response option.[16] If an included study fully adheres to the item, the answer to the item is judged as 'Yes' and this indicates the study has a good methodological quality in the aspect of such item. If an included study partially adheres to the item, the answer to the item is judged as 'Partial yes'. If an included study does not adhere to the item or provides no information to rate the item, the answer to the item is judged as 'No'.[16] We will not combine the items to create an overall score. Instead, the potential impact of each item on the results of this study will be considered separately.

Two reviewers (NS and CMF) will evaluate the methodological quality of reviews from a sample (10%) of eligible reviews and achieve good agreement (at least 80%), with the remainder extracted by one reviewer (NS).[16]

## Reviewers' training and double-checking for accuracy

The two assessors (NS and CMF) will pilot the forms for the data extraction and the methodological quality assessment using a set of 5% of the included reviews.[17] After the piloting round, the results will be discussed and, if necessary, the forms will be refined.

A third reviewer (MvdL) will independently check a random sample (10%) for the different study's phases (selection, extraction and quality assessment of the data) for accuracy. Any remaining disagreements will be discussed among all authors to reach a consensus decision.

## Statistical analysis

The overall prevalence of spin and the prevalence of each spin pattern presented in the included systematic or literature reviews will be described. We anticipate that the spin patterns identified in the included studies will vary with respect to their labels, definitions and classification systems used to organise and present them. If the classifications of patterns of spin (and consequently overall spin which is the sum of different patterns) vary only slightly across reviews, we will pool the reported overall prevalences and the prevalences of each spin pattern separately across reviews based on meta-analysis for single proportions with random effect models. If, however, the categories of spin patterns (and thus the definitions of overall spin) vary substantially across reviews, the overall prevalence of spin and the prevalence of spin patterns will be described qualitatively. When the meta-analysis is possibly performed, statistical heterogeneity of the meta-analysis will be assessed with $I^2$ test.[18] The heterogeneity is considered large if $I^2 > 50\%$. If it shows large heterogeneity and the number of included studies in a meta-analysis is >10, multivariate meta-regression analysis will be used to explore the possible factors for the heterogeneity. To assess the publication bias of the included studies, a funnel plot will be used when the number of the studies included in the meta-analysis is larger than 10.[18] The statistical analysis will be performed via R software V.3.3 (R Development Core Team).

## Assessment of the seriousness of the impact of spin patterns

The seriousness of the impact of spin patterns indicates how much consequence a certain spin pattern may cause in research and clinical practice if such spin pattern occurs in biomedical publications. We are planning to perform a Delphi consensus study to rate the seriousness of the spin patterns. The Delphi method is a structured process which uses a series of questionnaires or rounds to gather and to provide information in a panel of experts.[19] Compared with other possible study designs (eg, RCTs), the Delphi method is more simple to design and more

flexible to conduct.[20] Besides, it enables anonymity, which encourages experts' creativity, honesty, independent thinking and balanced consideration of ideas while reducing the risk of group dynamics negatively affecting the outcomes.[20 21] The experts' opinions are given equal weighting in the operation of a Delphi.[20 21] However, the Delphi method has been criticised as lacking objectivity and having problematic reliability, validity and credibility.[22] This is because the outcomes from the Delphi method are solely based on experts' opinions, rather than on the more objective evidence from research studies like RCTs.

In the study, we will invite experts in the field defined as authors of the included reviews. We will invite one author per included review, or two if the number of reviews is too small. We attempt to include a total of 10–15 experts. If the number of reviews is greater than 15, we will draw a random sample. We will set up an initial video conference to clarify the aim of the consensus study. For the consensus study, we will ask the experts to rate the seriousness of the impact of spin patterns on research and clinical practice using a 7-point rating scale in which '1' indicates that the spin pattern is not serious at all and '7' indicates that the spin pattern is extremely serious. In addition, and most importantly, the experts will also provide a written justification for the rating. The mean scores will be calculated and the range of the scores will be presented for each spin pattern. A researcher from the core team will present the results (ie, anonymous ratings, justifications and mean scores) to the individual group members in written form, ask them to review the results, and possibly adjust their own judgments based on the summary. We will repeat this process until no further changes occur. The final version will provide, for each spin pattern, the individual ratings (to show the level of agreement), the mean rating and a justification (if the ratings are heterogeneous, then we will provide explanation for the disagreement). It is not the aim that all members agree on a specific numerical rating and justification, that is, disagreement regarding the importance of some pattern is a possible result of the study. If necessary, for example, to discuss questions that are too complex for written communication, we may set up additional video conferences. If necessary, we may first develop a new classification for spin patterns together with the experts using similar methods (see the Potential development of a comprehensive classification scheme for spin patterns section).

## ETHICS AND DISSEMINATION
The present protocol has been approved by Academic Centre for Dentistry Amsterdam Ethics Review Committee (2020250). We will publish results in a peer-reviewed scientific journal.

**Author affiliations**
[1]Department of Oral Public Health, Academic Centre for Dentistry Amsterdam, Amsterdam, The Netherlands
[2]Department of Dentistry – Quality and Safety of Oral Healthcare, Radboud University Medical Center, Radboud Universiteit, Nijmegen, The Netherlands
[3]Section for Translational Health Economics, Medical Faculty, Heidelberg University, Heidelberg, Germany
[4]Department of Health Research Methods, Evidence, and Impact, McMaster University, Hamilton, Ontario, Canada
[5]Department of Clinical Research, Basel Intitute for Clinical Epidemiology and Biostatistics, University of Basel and University Hospital Basel, Basel, Switzerland
[6]Faculty of Dentistry, Department of Periodontology and Operative Dentistry, University Hospital Münster, Münster, Germany

**Acknowledgements** We gratefully acknowledged Mr René H.J. Otten, from University Library, Vrije Universiteit Amsterdam, the Netherlands for his assistance in the modification of the search strategies.

**Contributors** NS and CMF drafted the first version of the protocol and subsequently incorporated the suggested revisions. MvdL, GvdH, SL and SS commented and revised each section of the protocol.

**Funding** The authors have not declared a specific grant for this research from any funding agency in the public, commercial or not-for-profit sectors.

**Competing interests** None declared.

**Patient and public involvement** Patients and/or the public were not involved in the design, or conduct, or reporting or dissemination plans of this research.

**Patient consent for publication** Not required.

**Provenance and peer review** Not commissioned; externally peer reviewed.

**ORCID iDs**
Naichuan Su http://orcid.org/0000-0001-8034-9410
Stefan Schandelmaier http://orcid.org/0000-0002-8429-0337

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
