## [Reviewer comments · BMJ Open]

ARTICLE DETAILS

TITLE (PROVISIONAL)	Published patterns of spin in biomedical literature: A protocol for a meta-research study
AUTHORS	Su, Naichuan; van der Linden, Michiel; van der Heijden, Geert; Listl, Stefan; Schandelmaier, Stefan; Faggion, Clovis M.

VERSION 1 – REVIEW

REVIEWER	Bero, Lisa University of Sydney Faculty of Health Sciences, Pharmacy
REVIEW RETURNED	14-Oct-2020

GENERAL COMMENTS	The authors cite a previous systematic review of spin studies in their introduction (Chiu K, Grundy Q, Bero L (2017) 'Spin' in published biomedical literature: A methodological systematic review. PLoS Biol 15(9): e2002173. https://doi.org/10.1371/journal.pbio.2002173). This protocol appears to be a proposed replication or update of this study, with one exception – the section on Assessment of the seriousness of the impact of spin patterns (page 10, starting line 18). To my knowledge, a Delphi panel to rate the seriousness of spin patterns has not been conducted. I would refocus the protocol on this novel aspect. This protocol could update the 2017 PLoS Biol review cited above, but there are a number of methodological issues that would need to be resolved first: 1) Limiting the search to systematic reviews and literature reviews of spin will likely miss studies. For example, some studies of spin are cohort studies or cross-sectional studies.2) The term “spin pattern” is defined as including “labels, definitions and classification schemes.” (page 7, line 44) This will extract a limited amount of information about spin from the included studies. The previous review extracted information on how spin was defined, level of spin, practices used to spin results, and factors associated with spin.3) The previous review grouped these findings inductively to develop a categorization of spin. It is not clear in the current protocol how a different categorization of spin will be developed. The authors proposal (page 9, line 28) is “The team members will consider the different definitions and classifications and suggest whether the categories could be harmonized by lumping together and splitting up. If splitting up is necessary, we would have to re-assess the prevalence of patterns of spin in primary studies classified in categories that require splitting up. Therefore, we may have to approach the investigators of the original studies.”
---

	Furthermore, reassessing spin in primary studies will be problematic if the protocol authors use a classification that is different from the authors of the original studies. 4) Using AMSTAR-2 to assess risk of bias is not appropriate. AMSTAR is appropriate if only systematic reviews are included. As noted above, not all studies of spin are systematic reviews. 5) Studies of spin are very heterogeneous in terms of research topics and study designs included. In the previous review, we were not able to conduct a meta-analysis of prevalence of spin for this reason. It is unlikely that a quantitative synthesis will be possible.
--	--

REVIEWER	Koroleva, Anna Zurich Universities of Applied Sciences
REVIEW RETURNED	09-Nov-2020

GENERAL COMMENTS	The manuscript of Su et al. presents a protocol for a meta-research study of patterns of spin (unjustified positive reporting of research results). I. Boutron and colleagues first formulated the problem of spin in biomedical domain in 2010. Since then, several studies have shown high prevalence of spin in various medical domains and across different study designs. Spin in biomedical articles was demonstrated to have negative influence on clinical practice and media coverage. Su et al. claim that their study will focus on the prevalence of spin patterns and seriousness of the impact of spin patterns on research and clinical practice. Spin is an alarming problem, and it is important to study its prevalence and impact. In particular, the prevalence and impact of various types/patterns of spin has not been previously assessed. The proposed meta-research study is hence of high interest. However, the protocol has a few issues that need to be addressed to improve the protocol. I. Introduction 1) The authors make some claims about the impact of spin: p. 4 lines 12 – 13: “Such unjustifiable and misleading misrepresentation may flatter research and make it more attractive for publication and citation and consequently may receive unwarrantably higher impact scores.” p. 4 lines 22 – 24: “Spin may... stimulate ineffective or even harmful financial investments from funding organizations and health institutions.” It has been proved that spin impacts the clinicians’ perception of clinical study results and causes spin in media coverage. However, to my best knowledge, impact of spin on citations, impact scores and financial investments has not been studied. These claims need to be supported by a citation. If it is a hypothesis, it needs to be made clear. 2) p. 4 lines 43 – 45: “Those spin patterns can be classified into three main forms: misleading reporting, misleading interpretation, and inappropriate extrapolation (1).” The cited review (1) specifies 4 classes of spin: “(1) reporting practices that distort the interpretation of results and create misleading conclusions, suggesting a more favourable result; (2) discordance between results and their interpretation, with the interpretation being more favourable than the results; (3) attribution of causality when study design does not allow for it; and (4) overinterpretation or inappropriate extrapolation of results.” On the contrary, some earlier works introduce indeed three categories of spin listed by Su et al.:
---

Lazarus, C., Haneef, R., Ravaud, P. et al. Classification and prevalence of spin in abstracts of non-randomized studies evaluating an intervention. *BMC Med Res Methodol* 15, 85 (2015). <https://doi.org/10.1186/s12874-015-0079-x>

Yavchitz A, Ravaud P, Altman DG, Moher D, Hrobjartsson A, Lasserson T, Boutron I. A new classification of spin in systematic reviews and meta-analyses was developed and ranked according to the severity. *J Clin Epidemiol*. 2016 Jul;75:56-65. doi: 10.1016/j.jclinepi.2016.01.020. Epub 2016 Feb 2. PMID: 26845744.

Please use a correct reference.

3) p. 5 lines 37 – 41: “The prevalence of spin patterns in biomedical publications and seriousness of the impact of different spin patterns on research and clinical practice has not been assessed”

This claim seems to be too strong. The first study of spin (Boutron I, Dutton S, Ravaud P, Altman DG. Reporting and Interpretation of Randomized Controlled Trials With Statistically Nonsignificant Results for Primary Outcomes. *JAMA*. 2010;303(20):2058–2064. doi:10.1001/jama.2010.651) assessed the prevalence of different spin types/patterns. There are a few other studies assessing the prevalence of spin in various medical domains, e.g.:

- surgical research: P. S. Fleming. Evidence of spin in clinical trials in the surgical literature. *Ann Transl Med.*,4,19(385), Oct 2016. doi: 10.21037/atm.2016.08.23.

- cardiovascular diseases: M. Khan, N. Lateef, T. Siddiqi, K. Abdur Rehman, S. Alnaimat, S. Khan, H. Riaz, M. Hassan Murad, J. Mandrola, R. Doukky, and R. Krasuski. Level and prevalence of spin in published cardiovascular randomized clinical trial reports with statistically non-significant primary outcomes: A systematic review. *JAMA Network Open*, 2:e192622, 05 2019. doi:10.1001/jamanetworkopen.2019.2622

- cancer: F. E. Vera-Badillo, M. Napoleone, M. K. Krzyzanowska, S. M. Alibhai, A.-W. Chan, A. Ocana, B. Seruga, A. J. Templeton, E. Amir, and I. F. Tannock. Bias in reporting of randomised clinical trials in oncology. *European Journal of Cancer*, 61:29 – 35, 2016. ISSN 0959-8049. doi: 10.1016/j.ejca.2016.03.066.

- obesity: J. Austin, C. Smith, K. Natarajan, M. Som, C. Wayant, and M. Vassar. Evaluation of spin within abstracts in obesity randomized clinical trials: A cross-sectional review: Spin in obesity clinical trials. *Clinical Obesity*, 9:e12292, 12 2018. doi: 10.1111/cob.12292.)

- otolaryngology: C. M. Cooper, H. M. Gray, A. E. Ross, T. A. Hamilton, J. B. Downs, C. Wayant, and M. Vassar. Evaluation of spin in the abstracts of otolaryngology randomized controlled trials: Spin found in majority of clinical trials. *The Laryngoscope*, 12 2018. doi: 10.1002/lary.27750.

- anaesthesiology: N. Kinder, M. Weaver, C. Wayant, and M. Vassar. Presence of 'spin' in the abstracts and titles of anaesthesiology randomised controlled trials. *British Journal of Anaesthesia*, 122,11 2018. doi: 10.1016/j.bja.2018.10.023

- wound care: S. Lockyer, R. W. Hodgson, J. C. Dumville, and N. Cullum. "Spin" in wound care research: the reporting and interpretation of randomized controlled trials with statistically non-significant primary outcome results or unspecified primary outcomes. In *Trials*, 2013.

The impact of spin has also been assessed in previous studies. To justify the importance and to show the novelty of the proposed meta-research study, the authors should explain why studying the

prevalence and impact of spin patterns (vs. spin in general) is necessary and how it would add to the current knowledge.
4) p. 5 lines 44 – 46: “The overall aim of the present study is to derive recommendations for prevention of spin in future research.” The protocol seems to focus on assessing the prevalence of spin patterns and their impact on clinical practice and research. It would be good to clarify how this assessment could translate into recommendations for prevention of spin.

II. Methods and analysis

- 1) p. 6 line 31 – 32: “the snowballing method will be used to broaden the search (12)” – the reference 12 does not seem to mention snowballing. Please provide a correct reference.
- 2) p. 6 line 26: “... first publication on the topic spin (5)” – it would be more precise to say that this paper (5) was the first to identify “spin” in clinical trials. It is not the first paper on the problem of spin/distorted reporting (see the references of the paper 5).
- 3) p. 6 line 36: “to check the first 20 “similar studies”” – it would be of practical interest to clarify why authors decided to check this particular number (20) of studies. Was it set empirically? Is it the “best practice” in the field?
- 4) p. 7 lines 11 – 14: “Two reviewers (NS and CMF) will select a sample (10%) of eligible studies to achieve good agreement (at least 80%) on in- and exclusion of publications, and thereafter the remainder selected by one reviewer (NS) (14)” – why is 14 cited here?

In the subsection “Assessment of the seriousness of the impact of spin patterns”, it would be interesting to discuss the choice of Delphi study as a method of assessment. Previously, the impact of spin was assessed in randomized trials:

Boutron I, Altman DG, Hopewell S, et al. Impact of spin in the abstracts of articles reporting results of randomized controlled trials in the field of cancer: the SPIIN randomized controlled trial. *J Clin Oncol* 2014;32:4120-6

Boutron, I., Haneef, R., Yavchitz, A. et al. Three randomized controlled trials evaluating the impact of “spin” in health news stories reporting studies of pharmacologic treatments on patients'/caregivers' interpretation of treatment benefit. *BMC Med* 17, 105 (2019). <https://doi.org/10.1186/s12916-019-1330-9>

Conducting a randomized trial is a more difficult but a more objective way to assess the impact of spin. Preferring a Delphi study needs to be justified by discussing the advantages and drawbacks of both methods.

Minor remarks / typos:

- p.4 line 52 “second outcomes” – the commonly used term is “secondary outcomes”
- p. 5 lines 14 – 15 “of I patterns” – looks like a typo (“the patterns”?)
- p.8 line 32 “over score” – overall score?
- p. 8 line 46 “The two assessors (NS and CM)” – seems that the second assessor should be “CMF”, as above
- p. 9 line 25 “potentially classification schemes” – “potential classification schemes”

VERSION 1 – AUTHOR RESPONSE

Reviewer 1:

1. Limiting the search to systematic reviews and literature reviews of spin will likely miss studies. For example, some studies of spin are cohort studies or cross-sectional studies.

Response: We feel that reviews are the most appropriate to gain insight into the overall prevalence spin and the prevalence of spin patterns in medical publications by summarizing/pooling the multiple original primary studies. Based on our understanding, cohort studies or cross-sectional studies mainly focus on the individual level, rather than study level. So, review studies are likely to be more informative on spin of publications than classical cohort and cross-sectional studies.

Chiu et al. 2017 did not regard around 29% (10/35) of the studies they included in their systematic review as reviews. However, those 10 studies classified as non-reviews by Chiu et al. 2017 have the nature of systematic/literature reviews. Based on our understanding, those studies can be regarded as “review” studies because they analyzed research already conducted in primary sources and generally summarized the current state of research on a given topic based on the systematic methodologies. That is, those studies all had specific research questions, used certain search strategies to search the possible original studies, used certain inclusion/exclusion criteria to screen the original studies, reviewed and extracted data from the eligible primary studies, and synthesized the data quantitatively or qualitatively accordingly. So, they will be included in our present study as “review” studies.

The search strategy of the current protocol uses “systematic review”[publication type] as a mesh term, which indicates that even if the primary study per se did not use the word “review” in the title, abstract or main text, it is also very likely to be identified if the publication type is a review. Besides, to improve the comprehensiveness of search and identify potentially eligible studies which are not retrieved from the databases based on the search strategies, we will inspect the reference lists of the included primary studies (backward citation searching) and we will also inspect the first 20 “similar studies” for each included study. In this case, we expect that those studies which were regarded as cohort studies or cross-sectional studies in Chiu et al. 2017 can be identified.

To make the inclusion criteria of the present protocol clearer, the first criterium has been modified and shown as follows (see marked revision on page 7 line 204-211):

They reported systematic or literature reviews on spin. That is, the publications aimed to examine spin for the primary studies in a specific or broad field of medicine. The review articles are defined as a more or less systematic way of collecting and synthesizing previous research.¹ A review will be defined as systematic when authors of the review made clear the intention of performing a systematic review. If a publication met the definition based on its adopted methods without using the word “review” in the titles, abstracts, or main texts, it will also be included.

2. The term “spin pattern” is defined as including “labels, definitions and classification schemes.” (page 7, line 44) This will extract a limited amount of information about spin from the included studies. The previous review extracted information on how spin was defined, level of spin, practices used to spin results, and factors associated with spin.

Response: The present protocol focuses on the various spin patterns. That is, the prevalence of the spin patterns and the impact of the spin patterns on research will be assessed. The reason why we focus on the spin patterns is that we would like to show which spin patterns may occur most frequently in research and which spin patterns may have the most serious impact on research. The overall prevalence of spin in the medical publications was around 57% in both abstracts and main texts based on Chui et al.,² which is high. Based on findings of the present study, the researchers, clinicians, and policymakers can be aware of and get rid of those spin patterns when they read and write scientific papers. The increase of the awareness on spin (patterns) may help prevent spin in the future publications and prevent the use and transfer of the spinned evidence in research. Therefore,

we expect that the present study can help transfer of knowledge from research in a more reliable and accurate manner.

In the present study, we do not focus on the definitions of spin, level of spin, practices used to spin results, and factors associated with spin. This is because this does not fall in our scope and we think it is not very necessary to either repeat or update what Chiu et al. already did very well in 2017 only 3 years later.

We have emphasized the importance and necessities of the present study in the introduction section (see marked revision on page 5-6 line 149-166):

Several studies on the prevalence of spin and the impact of spin on research in general have been published ... These studies have revealed that spin is highly frequent in medical publications and can significantly impact both clinicians` and patients` decision-making in clinical practice. The main aim of these studies was to define, describe and understand spin. From the publications on spin available to date, different patterns of spin seem to emerge, but a comprehensive overview of spin patterns is lacking. Spin patterns vary in the prevalence in research and in the consequences on research. The ignorance on those spin patterns with the highest prevalence and the most serious impact on research may widely and severely harm the reliability, transparency, and accuracy of translation, dissemination, and implementation of the evidence from medical and health research to practice. Identification of the prevalence and the impact of spin patterns will help increase the awareness of researchers, clinicians, reviewers, and policymakers on which spin patterns should be given most attention when they read, review and write scientific papers. Besides, to develop the practical and targeted guidance on the identification and prevention of spin in research, the manifestations of spin should be known. The specific and targeted guidance on how to identify and prevent such spin patterns in research can be made accordingly to help prevent spin in the future publications and prevent the use and transfer of the spinned evidence in research.

3. The previous review grouped these findings inductively to develop a categorization of spin. It is not clear in the current protocol how a different categorization of spin will be developed. The authors proposal (page 9, line 28) is "The team members will consider the different definitions and classifications and suggest whether the categories could be harmonized by lumping together and splitting up. If splitting up is necessary, we would have to re-assess the prevalence of patterns of spin in primary studies classified in categories that require splitting up. Therefore, we may have to approach the investigators of the original studies." Furthermore, reassessing spin in primary studies will be problematic if the protocol authors use a classification that is different from the authors of the original studies.

Response: We agree that it is preferable to adhere to existing definitions and classifications, rather than to develop new categorizations. In the modified version of the protocol, we have therefore decided to delete the section "potential development of a comprehensive classification scheme for spin patterns" because similar work has been done by Chiu et al. 2017² and due to the lack of feasibility on the re-categorization of spin patterns in primary studies.

4. Using AMSTAR-2 to assess risk of bias is not appropriate. AMSTAR is appropriate if only systematic reviews are included. As noted above, not all studies of spin are systematic reviews.

Response: In the present protocol, we will only include systematic or literature reviews (see the response under item 1). We acknowledge that the AMSTAR-2 is mainly used to assess the methodological quality of the systematic reviews with health interventions. However, the systematic reviews or literature reviews included in the present study are methodological on spin in the included primary studies rather than the health interventions. But so far, as far as we know, there is no validated tool assessing the methodological reviews. Therefore, in the present study, we only selected 8 items from the AMSTAR-2 which are applicable and relevant for all types of systematic or literature reviews.

5. Studies of spin are very heterogeneous in terms of research topics and study designs included. In the previous review, we were not able to conduct a meta-analysis of prevalence of spin for this reason. It is unlikely that a quantitative synthesis will be possible.

Response: Thanks for sharing this important finding with us. In the statistical analysis section, we have emphasized that if the categories of spin patterns and the definitions of overall spin vary substantially across the included reviews, the overall prevalence of spin and the prevalence of spin patterns will be described separately. The modified text is shown below (see marked revision on page 10 line 316-319):

If, however, the categories of spin patterns (and thus the definitions of overall spin) vary substantially across reviews, the overall prevalence of spin and the prevalence of spin patterns will be described qualitatively.

Reviewer 2

1. Introduction: The authors make some claims about the impact of spin:

p. 4 lines 12 – 13: “Such unjustifiable and misleading misrepresentation may flatter research and make it more attractive for publication and citation and consequently may receive unwarrantably higher impact scores.”

p. 4 lines 22 – 24: “Spin may... stimulate ineffective or even harmful financial investments from funding organizations and health institutions.”

It has been proved that spin impacts the clinicians’ perception of clinical study results and causes spin in media coverage. However, to my best knowledge, impact of spin on citations, impact scores and financial investments has not been studied. These claims need to be supported by a citation. If it is a hypothesis, it needs to be made clear.

Response: The two sentences in the introduction section have been modified based on the reviewer’s suggestion. The modified sentences are shown below (see marked revision on page 4 line 102 and 108):

It can be hypothesized that such unjustifiable and misleading misrepresentation may flatter research and make it more attractive for publication and citation and consequently may receive unwarrantably higher impact scores.

Based on the hypothesis, spin may lead to overpromising and misleading information in transfer of knowledge, introduce misconceptions to researchers, clinicians, and policymakers, and stimulate ineffective or even harmful financial investments from funding organizations and health institutions.

2. p. 4 lines 43 – 45: “Those spin patterns can be classified into three main forms: misleading reporting, misleading interpretation, and inappropriate extrapolation (1).”

The cited review (1) specifies 4 classes of spin: “(1) reporting practices that distort the interpretation of results and create misleading conclusions, suggesting a more favourable result; (2) discordance between results and their interpretation, with the interpretation being more favourable than the results; (3) attribution of causality when study design does not allow for it; and (4) overinterpretation or inappropriate extrapolation of results.”

On the contrary, some earlier works introduce indeed three categories of spin listed by Su et al.: Lazarus, C., Haneef, R., Ravaud, P. et al. Classification and prevalence of spin in abstracts of non-randomized studies evaluating an intervention. *BMC Med Res Methodol* 15, 85 (2015). <https://doi.org/10.1186/s12874-015-0079-x>

Yavchitz A, Ravaud P, Altman DG, Moher D, Hrobjartsson A, Lasserson T, Boutron I. A new classification of spin in systematic reviews and meta-analyses was developed and ranked according to the severity. *J Clin Epidemiol*. 2016 Jul;75:56-65. doi: 10.1016/j.jclinepi.2016.01.020. Epub 2016 Feb 2. PMID: 26845744.

Please use a correct reference.

Response: The reference has been corrected (see marked revision on page 4 line 123).

3. p. 5 lines 37 – 41: “The prevalence of spin patterns in biomedical publications and seriousness of the impact of different spin patterns on research and clinical practice has not been assessed”

This claim seems to be too strong. The first study of spin (Boutron I, Dutton S, Ravaud P, Altman DG. Reporting and Interpretation of Randomized Controlled Trials With Statistically Nonsignificant Results for Primary Outcomes. *JAMA*. 2010;303(20):2058–2064. doi:10.1001/jama.2010.651) assessed the prevalence of different spin types/patterns. There are a few other studies assessing the prevalence of spin in various medical domains, e.g.:

- surgical research: P. S. Fleming. Evidence of spin in clinical trials in the surgical literature. *Ann Transl Med.*,4,19(385), Oct 2016. doi: 10.21037/atm.2016.08.23.
- cardiovascular diseases: M. Khan, N. Lateef, T. Siddiqi, K. Abdur Rehman, S. Alnaimat, S. Khan, H. Riaz, M. Hassan Murad, J. Mandrola, R. Doukky, and R. Krasuski. Level and prevalence of spin in published cardiovascular randomized clinical trial reports with statistically non-significant primary outcomes: A systematic review. *JAMA Network Open*, 2:e192622, 05 2019. doi:10.1001/jamanetworkopen.2019.2622
- cancer: F. E. Vera-Badillo, M. Napoleone, M. K. Krzyzanowska, S. M. Alibhai, A.-W. Chan, A. Ocana, B. Seruga, A. J. Templeton, E. Amir, and I. F. Tannock. Bias in reporting of randomised clinical trials in oncology. *European Journal of Cancer*, 61:29 – 35, 2016. ISSN 0959-8049. doi: 10.1016/j.ejca.2016.03.066.
- obesity: J. Austin, C. Smith, K. Natarajan, M. Som, C. Wayant, and M. Vassar. Evaluation of spin within abstracts in obesity randomized clinical trials: A cross-sectional review: Spin in obesity clinical trials. *Clinical Obesity*, 9:e12292, 12 2018. doi: 10.1111/cob.12292.)
- otolaryngology: C. M. Cooper, H. M. Gray, A. E. Ross, T. A. Hamilton, J. B. Downs, C. Wayant, and M. Vassar. Evaluation of spin in the abstracts of otolaryngology randomized controlled trials: Spin found in majority of clinical trials. *The Laryngoscope*, 12 2018. doi: 10.1002/lary.27750.
- anaesthesiology: N. Kinder, M. Weaver, C. Wayant, and M. Vassar. Presence of 'spin' in the abstracts and titles of anaesthesiology randomised controlled trials. *British Journal of Anaesthesia*, 122,11 2018. doi: 10.1016/j.bja.2018.10.023
- wound care: S. Lockyer, R. W. Hodgson, J. C. Dumville, and N. Cullum. "Spin" in wound care research: the reporting and interpretation of randomized controlled trials with statistically non-significant primary outcome results or unspecified primary outcomes. In *Trials*, 2013.

Response: Sorry for the confusion caused by this inaccuracy. Actually, our intention is to show that the relevant meta-research studies (rather than the original spin studies) on the spin pattern is scarce. The sentence has been modified and shown below (see marked revision on page 6 line 167-169):

To date, however, meta-research studies on the prevalence of spin patterns in biomedical publications and the impact of different spin patterns on research and clinical practice are scarce.

4. The impact of spin has also been assessed in previous studies. To justify the importance and to show the novelty of the proposed meta-research study, the authors should explain why studying the prevalence and impact of spin patterns (vs. spin in general) is necessary and how it would add to the current knowledge.

Response: The importance and necessities of the assessment of spin patterns in the present study has been stated in the introduction section. It is shown below (see page 5-6 line 149-166):

Several studies on the prevalence of spin and the impact of spin on research in general have been published ... These studies have revealed that spin is highly frequent in medical publications and can significantly impact both clinicians` and patients` decision-making in clinical practice. The main aim of these studies was to define, describe and understand spin. From the publications on spin available to date, different patterns of spin seem to emerge, but a comprehensive overview of spin patterns is lacking. Spin patterns vary in the prevalence in research and in the consequences on research. The ignorance on those spin patterns with the highest prevalence and the most serious impact on research may widely and severely harm the reliability, transparency, and accuracy of translation,

dissemination, and implementation of the evidence from medical and health research to practice. Identification of the prevalence and the impact of spin pattern will help increase the awareness of researchers, clinicians, reviewers, and policymakers on which spin patterns should be given most attention when they read, review and write scientific papers. Besides, to develop the practical and targeted guidance on the identification and prevention of spin in research, the manifestations of spin should be known. The specific and targeted guidance on how to identify and prevent such spin patterns in research can be made accordingly to help prevent spin in the future publications and prevent the use and transfer of the spinned evidence in research.

5. p. 5 lines 44 – 46: “The overall aim of the present study is to derive recommendations for prevention of spin in future research.”

The protocol seems to focus on assessing the prevalence of spin patterns and their impact on clinical practice and research. It would be good to clarify how this assessment could translate into recommendations for prevention of spin.

Response: The idea of the study is to identify which spin patterns occur most frequently and which spin patterns have the most serious consequences on research in biomedical publications. Then, we will develop the specific and targeted guidance/suggestions on how to identify and prevent those specific spin patterns with high prevalence or serious consequence on research for clinicians, researchers, and policymakers when they read and write scientific papers.

The aim of the study (the last paragraph of the introduction section) has been modified (see marked revision on page 6 line 171-178):

Therefore, the aims of the study are to (1) identify reported spin patterns and assess the prevalence of spin in general and the prevalence of spin patterns reported in biomedical literature based on previously published systematic reviews and literature reviews on spin; and (2) assess the seriousness of the impact of spin patterns on research and clinical practice. Based on the spin patterns found in the current study which are most frequent in research or have the most serious consequences on research, we will derive recommendations for researchers, clinicians, and policymakers to prevent such spin patterns in future research.

6. II. Methods and analysis

- 1) p. 6 line 31 – 32: “the snowballing method will be used to broaden the search (12)” – the reference 12 does not seem to mention snowballing. Please provide a correct reference.

Response: The reference has been corrected (see marked revision on page 7 line 196). The new reference is shown below: Greenhalgh T, Peacock R. Effectiveness and efficiency of search methods in systematic reviews of complex evidence: audit of primary sources. *BMJ* 2005;331:1064-5.

7. p. 6 line 26: “... first publication on the topic spin (5)” – it would be more precise to say that this paper (5) was the first to identify “spin” in clinical trials. It is not the first paper on the problem of spin/distorted reporting (see the references of the paper 5).

Response: The sentence has been modified accordingly and shown below (see marked revision on page 6 line 191-192):

Searches will be limited from the last 10 years (from 2010), as from the first publication to identify spin in clinical trials.

8. p. 6 line 36: “to check the first 20 “similar studies”” – it would be of practical interest to clarify why authors decided to check this particular number (20) of studies. Was it set empirically? Is it the “best practice” in the field?

Response: Since to date we are unaware of any empirical evidence or best practice from the previous literature, the number “20” was determined mainly based on our own practical experience. We will comprehensively search the literature using citation tracking to assure that all the eligible studies will be identified. For this we will identify the first 20 most “similar studies” in PubMed for each included

primary study. These will be displayed in ranked order from most to least relevant (<https://pubmed.ncbi.nlm.nih.gov/help/>).

9. p. 7 lines 11 – 14: “Two reviewers (NS and CMF) will select a sample (10%) of eligible studies to achieve good agreement (at least 80%) on in- and exclusion of publications, and thereafter the remainder selected by one reviewer (NS) (14)” – why is 14 cited here?

Response: The reason why we cited the reference 14 here is that the AMSTAR2 guidance document (Online Appendix 1) of this reference reported that if one individual carried out selection of all studies, with a second reviewer checking agreement on a sample of studies, it is recommended that a Kappa score of 0.80 or greater should have been achieved. In the present study, we used the same approach and same cut-off for the good agreement for the inclusion and exclusion of the studies as what has been recommended in the reference. Hence, we consider this an approach which can be considered as sufficiently evidence-based for our aims.

10. In the subsection “Assessment of the seriousness of the impact of spin patterns”, it would be interesting to discuss the choice of Delphi study as a method of assessment. Previously, the impact of spin was assessed in randomized trials:

Boutron I, Altman DG, Hopewell S, et al. Impact of spin in the abstracts of articles reporting results of randomized controlled trials in the field of cancer: the SPIIN randomized controlled trial. *J Clin Oncol* 2014;32:4120-6
Boutron, I., Haneef, R., Yavchitz, A. et al. Three randomized controlled trials evaluating the impact of “spin” in health news stories reporting studies of pharmacologic treatments on patients’/caregivers’ interpretation of treatment benefit. *BMC Med* 17, 105 (2019). <https://doi.org/10.1186/s12916-019-1330-9>

Conducting a randomized trial is a more difficult but a more objective way to assess the impact of spin. Preferring a Delphi study needs to be justified by discussing the advantages and drawbacks of both methods.

Response: In the “Assessment of the seriousness of impact of spin patterns”, we have described the advantages and disadvantages of the Delphi method and compared it with other study designs (including RCT) briefly. The text is shown below (see marked revision on page 11 line 332-343):
The Delphi method is a structured process which uses a series of questionnaires or rounds to gather and to provide information in a panel of experts.³ Compared with other possible study designs (e.g. RCTs), the Delphi method is more simple to design and more flexible to conduct.⁴ Besides, it enables anonymity, which encourages experts` creativity, honesty, independent thinking, and balanced consideration of ideas while reducing the risk of group dynamics negatively affecting the outcomes.⁴
⁵ The experts` opinions are given equal weighting in the operation of a Delphi.^{4 5} However, the Delphi method has been criticized as lacking objectivity and having problematic reliability, validity, and credibility.⁶ This is because the outcomes from the Delphi method are solely based on experts` opinions, rather than on the more objective evidence from research studies like RCTs.

11. Minor remarks / typos:

p.4 line 52 “second outcomes” – the commonly used term is “secondary outcomes”

Response: “second outcomes” has been corrected to “secondary outcomes” (see marked revision on page 4 line 127).

12. p. 5 lines 14 – 15 “of I patterns” – looks like a typo (“the patterns”?)

Response: This sentence has been deleted from the manuscript due to the comment from the other reviewer.

13. p.8 line 32 “over score”– overall score?

Response: “over score” has been corrected to “overall score” (see marked revision on page 9 line 265).

14. p. 8 line 46 “The two assessors (NS and CM)” – seems that the second assessor should be “CMF”.

Response: “CM” has been corrected to “CMF” (see marked revision on page 9 line 273 and page 12 line 376).

15. p. 9 line 25 “potentially classification schemes” – “potential classification schemes”

Response: This sentence has been deleted from the manuscript due to the comment from the other reviewer.

References

1. Snyder H. Literature review as a research methodology: An overview and guidelines. J Bus Res 2019;104:333-9.
2. Chiu K, Grundy Q, Bero L. “Spin” in published biomedical literature: A methodological systematic review. PLoS Biol 2017;15:e2002173.
3. Dalkey N, Helmer O. An experimental application of the Delphi methods to the use of experts. Manage Sci 1963;9:458-67.
4. Vernon W. The Delphi technique: A review. Int J Ther Rehabil 2009;16:69-76.
5. Donohoe HM, Needham RD. Moving best practice forward: Delphi characteristics, advantages, potential problems, and solutions. Int J Tour Res 2009;11:415-37.
6. Keeney S, Hasson F, McKenna HP. A critical review of the Delphi technique as a research methodology for nursing. Int J Nurs Stud 2001;38:195-200.

VERSION 2 – REVIEW

REVIEWER	Bero, Lisa University of Sydney Faculty of Health Sciences, Pharmacy
REVIEW RETURNED	19-Feb-2021

GENERAL COMMENTS	Although the authors have defined more clearly what they mean by "systematic review," I still have concerns about limiting the search to systematic reviews, as some meta-studies of spin are not reviews. This is also a limitation in applying the AMSTAR tool to all the included studies, which may be of different designs.
--

REVIEWER	Koroleva, Anna Zurich Universities of Applied Sciences
REVIEW RETURNED	09-Mar-2021

GENERAL COMMENTS	The authors have carefully addressed my concerns and answered my questions. The protocol defines an interesting study of an alarming problem (spin), and has a potential to increase the community's awareness of the problem. I would like to thank the authors for their work on this manuscript, and consider the manuscript advisable for publication.
--

VERSION 2 – AUTHOR RESPONSE

Reviewer 1

1. Although the authors have defined more clearly what they mean by "systematic review," I still have concerns about limiting the search to systematic reviews, as some meta-studies of spin

are not reviews. This is also a limitation in applying the AMSTAR tool to all the included studies, which may be of different designs.

Response: We have added the points mentioned by the reviewer as the potential limitations in the protocol ((see the marked version on page 3 line 14-18):

- Limiting the search to literature reviews or systematic reviews only is a potential limitation of the study because some meta-studies may not be reviews.
- Applying the AMSTAR-2 checklist to all the included studies for methodological quality assessment is another potential limitation because some of the included studies may have other designs than systematic reviews.